# Smoking prevalence, core/periphery network positions, and peer influence: Findings from five datasets on US adolescents and young adults

Cheng Wang ⬤ *

Department of Sociology, Wayne State University, Detroit, MI, United States of America

* chengwang@wayne.edu

## Abstract

Smoking prevalence has decreased significantly among US adolescents and young adults in the past 20 years. It is possible that adolescent and young adult smokers were moving from core to peripheral positions in social networks and thus less influential as suggested in previous research on adult smokers. We construct five sample datasets to test these hypotheses but none of them receives much support. When the proportion of smokers is relatively higher in two sample datasets, smokers tended to be at more marginal network positions than nonsmokers, both smokers and nonsmoker could exert peer influence, and the magnitude of peer influence from smokers was even greater than that from nonsmokers. When smoking was less frequent in the other three sample datasets, smokers and nonsmokers were at random network positions and no peer influence on smoking behavior was detected. Therefore, core/periphery network positions are still the key linking smoking prevalence and peer influence among US adolescents and young adults but operating through a different mechanism from their adult counterparts. When scientists design and conduct prevention programs against adolescent and young adult smoking behavior, core/periphery network positions, smoking prevalence, and peer influence should all be taken into consideration.

## Introduction

Health researchers, policymakers, experts, and other stakeholders in the field have done a good job in controlling the smoking behavior among US adolescents and young adults. In the past 25 years the smoking prevalence among 8th, 10th, and 12th graders in the United States, as shown in Fig 1, is observed to increase at the beginning of 1990s, peak at mid-1990s, and then decline significantly in the next two decades [1].

In their classic work on the association between substance use and social networks, Christakis and Fowler [2] traced 12,067 participants aged 21 years old and above in the Framingham Heart Study from 1971 to 2003. They found that the smoking prevalence in their sample declined from 52% to 13% in the 32 years, the trend of which approximately matched that of

**Data Availability Statement:** This research uses data from Add Health, a program project directed by Kathleen Mullan Harris and designed by J. Richard Udry, Peter S. Bearman, and Kathleen

Mullan Harris at the University of North Carolina at Chapel Hill. Information on how to obtain the AddHealth data files is available on the AddHealth website (http://www.cpc.unc.edu/addhealth). The NetSense dataset is de-identified and attached as S1 File. The NetHealth data are publicly accessible via the following URL: http://sites.nd.edu/nethealth/data-2/.

**Funding:** The author received no specific funding for this work.

**Competing interests:** The authors have declared that no competing interests exist.

smoking prevalence among US adolescents and young adults. Moreover, smokers were more likely to occupy core positions and form a central and densely connected cluster in the social network when smokers outnumbered nonsmokers, e.g., in 1971, and those participants who smoked gradually moved to the peripheral positions of the network with just a few links or no connections at all when most people quit smoking, e.g., by 2000 [2].

However, the research on US adolescent and young adult smoking tells a story that doesn't quite match Christakis and Fowler [2]. This line of literature applies three categories of network positions defined in Ennett and Bauman [3]: a group or clique member whose friends are densely connected, a liaison or bridge whose friends are loosely connected, and an isolate who has only one or no friend. In one cross-sectional study, Henry and Kobus [4] used data collected from seven K-8 elementary schools in Chicago along with four K-8 elementary schools and three 6–8 junior high schools near Chicago in the late 1990s. The average smoking prevalence was 17%. They found that students whose friends were loosely connected, or liaisons, had higher odds of tobacco use than group members and isolates [4]. In another longitudinal study, Ennett et al. [5] used egocentric social network data collected from 13 schools in three public school districts of North Carolina between 2002 and 2004. The smoking prevalence was below 10% for students aged 11 and 12, between 10% and 20% for those aged 13 and 14, and between 20% and 30% for those aged 15 and 16. They found that students whose friends were densely connected, or group members, had lower odds of recent smoking at age 15; and they also found that isolates tended to report more recent smoking than group members at age 13 [5].

Tobacco control researchers are particularly interested in young adults from 18 to 24 years of age who are separated from adolescents aged 12–17 [6–9]. Based on Christakis and Fowler

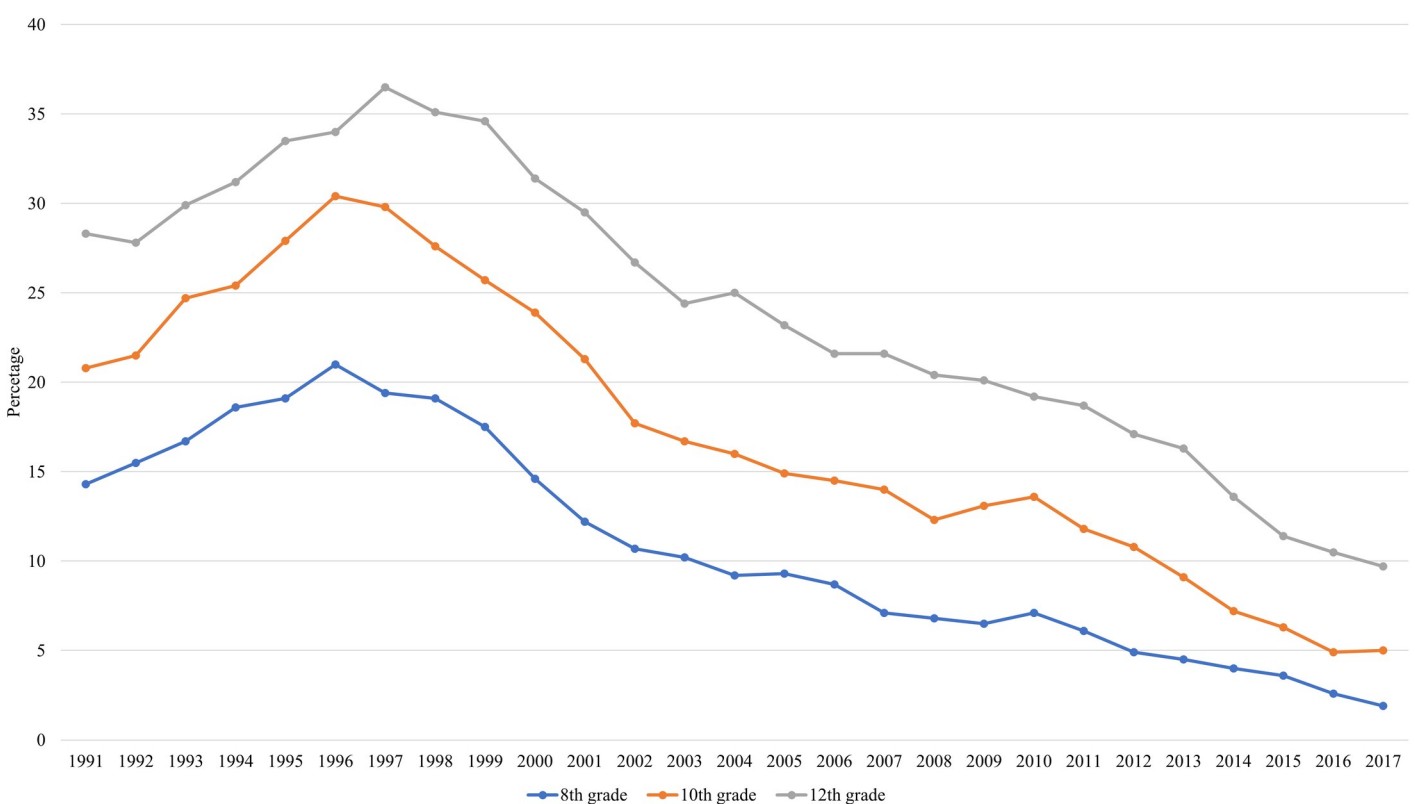

**Fig 1. 30-day prevalence of cigarette use, by grade, 1991–2017.** This figure is generated from statistics reported in Johnston et al. [1].

[2] focusing on adult participants, in this study we derive two hypotheses from their work, both of which we expect to be challenged in light of the research on US adolescent and young adult smoking:

I. Adolescent and young adult smokers are more likely to be at core of a social network when smoking is more prevalent; and

II. Adolescent and young adult smokers are more likely to be at margin of a social network when smoking is less prevalent.

Now we move on to the next research question, i.e., whether the magnitudes of peer influence from smokers and nonsmokers change in an environment with lower smoking prevalence. Peer influence refers to the propensity of a person to alter his or her behavior or attitude such that it becomes more similar to that of his or her contacts. While the smokers can influence their non-smoking contacts to start smoking, the nonsmokers can also influence their smoking contacts to quit smoking. Christakis and Fowler [2] found the person-to-person spread of smoking cessation helped reduce the smoking prevalence among adult participants in the Framingham Heart study, implying that the nonsmokers moving to the core of social network became more influential than smokers over the 32 years. A recent systematic review also confirmed the existence of peer influence in 15 out of 16 studies on US adolescent and young adult smoking [10]. However, there is limited research evaluating the magnitudes of peer influence of smokers and nonsmokers. In one study, Hass and Schaefer [11] found that high school students were more likely to be affected by their peers to start smoking than to quit smoking when smoking prevalence among US adolescents and young adults reached its peak in the mid-1990s.

The minority influence theory is appropriate to interpret the change in magnitude of peer influence. It assumed that majority and minority in a population, or the greater (more than half) subset of individuals and the smaller (less than half) subset of individuals, can both be influential, and the influence from majority on internalized attitude changes usually operates via the normative conformity process by core members in a context along the central route, while that from minority operates via the persuasion process by marginal members along the peripheral route [12]. Moreover, according to minority influence theory, the evolution of the Zeitgeist or social norms can drive the subsets of individuals to switch between majority and minority in the population [13, 14], and the size of minority and the magnitude of influence they exert usually go hand in hand [15]. This body of literature motivates our third hypothesis:

III. Adolescent and young adult smokers diminish their influence when they move from core to peripheral position of a social network because (a) the Zeitgeist is unfavorable to them and (b) their subset size decreases.

In this study we test these three hypotheses with five sample datasets. Three datasets focusing on US adolescent come from early waves of the National Longitudinal Study of Adolescent to Adult Health (Add Health) project [16], a time when smoking prevalence reached its peak among US adolescents and young adults. The other two datasets focusing on US young adults were from the NetSense project [17] and the NetHealth project [18, 19] conducted more recently at the University of Notre Dame, a time when smoking was less favorable. The adult network in the Framingham Heart study is majorly comprised of spouse and sibling ties which were relatively stable over the 32 years, with friendships accounting for only about 6% of all ties [2]. However, the adolescent networks in the Add Health study [16] and the young adult network in the NetSense project [17] and the NetHealth project [18, 19] are comprised of ties to peers. This could cause Christakis and Fowler's findings on adults to be different from that of adolescents and young adults in this study. In addition, social psychologists who advanced and enriched the minority influence theory basically relied on lab experiments instead of real-world data to reach their conclusions and largely ignored the network relationships and

contexts among their subjects. Therefore, in this study we aim to extend this line of research with several empirical datasets and present a more complete theory with regard to adolescent and young adult smoking, core and periphery network positions, and peer influence. We hope this study can offer insight into the interdependent mechanism of these three factors, which further provides some guidance for prevention scientists to design and conduct intervention programs against smoking behavior among targeted adolescents and young adults with various levels of cigarette/tobacco use prevalence, compositions of core and periphery network positions, and magnitudes of peer influence.

## Materials and methods

### Source data

The Add Health project [16] was approved by the Institutional Review Board at the University of North Carolina, School of Public Health. The NetSense project [17] and NetHealth project [18, 19] were approved by the Institutional Review Board at the University of Notre Dame.

Informed consent was given by the Add Health participants (or next of kin/caregiver) for their answers to be used in secondary data analysis. Written informed consents were obtained from all NetSense and NetHealth participants in the study.

**Sample datasets 1–3.** The first three sample datasets come from the well-known Add Health project [16], a large study of 132 high schools in the United States. The students in participating schools had an average smoking prevalence of 25%. They nominated up to five female and five male friends during the wave 1 In-Home survey conducted in 1995 and the wave 2 In-Home survey conducted in 1996. All students in 16 out of 132 schools were surveyed and thus had information on their complete friendship networks. Due to administrative errors, in one saturated school (i.e., school 175) the student IDs at earlier wave could not match those at later wave and among the other 15 saturated schools some students were only allowed to nominate one female and one male best friends during the wave 1 In-Home survey. After excluding school 175, we construct sample datasets from three schools in which at least 85% students didn't suffer from the second administrative error of limited nominations. The first school 58 was a rural public high school in the Midwest referred to as "Jefferson High" [20] and the substance use levels of its students ranked among the top 5 out of 132 participating schools. The second school 77 was a suburban public high school in the West referred to as "Sunshine High" [21]. The third school 28 was an urban private high school in the South. We only include the participants who showed up at both waves and thus the sample sizes for these three datasets are 631, 1166, and 125, respectively.

**Sample datasets 4–5.** The fourth and fifth sample datasets come from two case studies. Researchers at the University of Notre Dame randomly selected two cohorts of undergraduate students via the NetSense project [17] and the NetHealth project [18, 19]. The NetSense project recruited 196 participants in August 2011, and the NetHealth project recruited its first tier of 387 participants in August 2015 and the second tier of 96 participants in October 2015, which add up to 483 participants. They signed informed consent and periodically backed up call detailed records (CDRs) from their smartphones to a server administered by the project teams. The longitudinal social network data in the fourth sample dataset were constructed by aggregating the smartphone communication events among the 196 NetSense participants during fall semester 2011 and spring semester 2012, and that in the fifth sample dataset were constructed in the same way among 483 NetHealth participants during fall semester 2015 and spring semester 2016. The information on smoking frequency was retrieved from on-line surveys conducted during the corresponding semesters.

All five sample datasets contain the information on their participants' self-reported personal characteristics (i.e., gender, age, race, parental education level) and smoking levels at two consecutive time points. There are also nonnegligible differences across the datasets. First, the first three sample datasets were collected from high school students, while sample datasets 4–5 were collected from college students. Second, the NetSense and NetHealth projects used a different strategy from the Add Health study to collect information on friendship networks. The Add Health study used the name generator, i.e., a survey question asking each participant to recall the names of his or her friends. Instead, the NetSense and NetHealth projects built friendship networks among their participants with real-word communication events unobtrusively collected by their smartphones. Third, the Zeitgeist toward smoking in 1990s, when the first three sample datasets were collected, was different from that in 2010s, when sample datasets 4–5 were collected.

## Smoking measures

In sample datasets 1–3 participants were asked "During the past 30 days, on how many days did you smoke cigarettes?" at both waves and thus their responses ranged from 0 to 30. In sample datasets 4–5 participants were asked how often they used tobacco during each time window. Following previous research [22–24], we recode the response categories into four levels on a 30-day basis so that they are comparable across datasets: 0 = never, 1 = 1–3 days, 2 = 4–21 days, 3 = 22 or more days.

## Network measures

To evaluate the core/periphery network position of each participant, Christakis and Fowler [2] used degree centrality and eigenvector centrality, both of which are good candidates of coreness measures [25]. Degree centrality is the number of contact(s) an individual connects to [26] and individuals with low degrees are clearly peripheral in a network [27]. Eigenvector centrality is the principal eigenvector of a network adjacency matrix and an individual has higher value of eigenvector centrality when he or she connects to other well-connected contacts [28]. Degree centrality and eigenvector centrality have been used as indicators of core-periphery positions in previous research on lobbyist networks [29] and international investment networks [30]. In this study we include two additional measures. Betweenness centrality indicates the frequency of each individual to be on the shortest path between other individuals [26]. Though Borgatti and Everett [25] indicate betweenness centrality is not a good coreness measure, it reflects an individual's influence on spreading information, ideas, and behaviors in the network [31–33] as stated in Hypothesis III. For example, betweenness centrality is found to be highly associated with peer influence in crime co-offending networks [34] and online social networks [35]. Finally, we include a *k*-core measure. A *k*-core is a subgraph in which each node is adjacent to at least a minimum number, *k*, of the other nodes in the subgraph [36]. Individuals with high *k*-core values are more central in the network. *k*-core has been used as a coreness measure to study HIV risk [37, 38] and COVID-19 immunity [39].

## Analytical strategy

To test the first and second hypotheses, we estimate a structural equation model (SEM) for each sample dataset using Stata V16.0. The SEM analysis is appropriate for longitudinal study of the multiple-path relationships between adolescent/young adult peer networks and their smoking behavior [40, 41]. In this study one smoking measure and four network measures at the first wave serve as the independent variables and those at the later wave serve as the dependent variables. SEM can estimate unbiased standardized path parameters between multiple

causes and multiple outcomes simultaneously [42]. A comparative fit index (CFI) of value .95 or greater and a root mean square error of approximation (RMSEA) of value .06 or less are applied to evaluate the model fit [43], and both indices are found to be robust to the sample size biases [44]. The paths of particular interest are those between smoking behavior and network measures.

The Stochastic Actor-Based (SAB) modeling strategy [45, 46] implemented in RSiena [47] is applied to test Hypothesis III. The SAB modeling strategy has been widely used to investigate the peer influence via social networks on adolescent and young adult smoking behavior in recent years [10, 11, 21–24]. The average similarity effect indicates whether peer influence exists in an environment or not. The creation function is used to measure the magnitude of peer influence on increasing smoking levels and the endowment function is used to measure the magnitude of peer influence on decreasing smoking levels [11]. Given demographic variance in the five sample dataset, we also control for covariates such as gender, age, parental education, and race in the SAB models. We assess satisfactory model convergence with criteria of $t$ statistics for deviations from targets (i.e., ideally less than 0.10 for each parameter) and the overall maximum convergence ratio (i.e., ideally less than 0.25) [47].

## Results

### Smoking statistics

The smoking frequency in the five sample datasets is shown in Table 1. The proportion of smokers was about 50%-56% in sample dataset 1 and 22%-27% in sample dataset 2, respectively. Moreover, participants in sample dataset 1 showed a bimodal distribution at each end of the smoking frequency axes. As for the other three sample datasets, most participants were

**Table 1. Smoking levels and covariates from the five sample datasets.**

| Sample dataset | 1 | | 2 | | 3 | | 4 | | 5 | |
|---|---|---|---|---|---|---|---|---|---|---|
| | Wave1 | Wave2 | Wave1 | Wave2 | Wave1 | Wave2 | Wave1 | Wave2 | Wave1 | Wave2 |
| Smoking (past 30 days, %) | | | | | | | | | | |
| 0 = Never | 50% | 44% | 78% | 73% | 89% | 83% | 91% | 93% | 85% | 89% |
| 1 = 1–3 days | 10% | 10% | 7% | 10% | 2% | 8% | 8% | 5% | 2% | 3% |
| 2 = 4–21 days | 13% | 12% | 8% | 9% | 3% | 3% | 1% | 1% | 3% | 3% |
| 3 = 22 or more days | 27% | 34% | 7% | 8% | 6% | 6% | 0% | 1% | 10% | 5% |
| | Wave1 | | Wave1 | | Wave1 | | Wave1 | | Wave1 | |
| Age (mean, SD, range) | 15.53, 1.08, 14–18 | | 15.95, 1.01, 12–19 | | 14.19, 1.58, 11–18 | | 18.28, 0.48, 17–19 | | 18.01, 0.44, 17–20 | |
| Female (1 = Yes, %) | 48% | | 50% | | 64% | | 46% | | 48% | |
| Race (%) | | | | | | | | | | |
| White | 96% | | 9% | | 49% | | 68% | | 65% | |
| Latino | 1% | | 39% | | 7% | | 10% | | 14% | |
| Black | 0% | | 19% | | 40% | | 6% | | 6% | |
| Asian | 1% | | 32% | | 1% | | 12% | | 9% | |
| Other race | 2% | | 1% | | 3% | | 4% | | 6% | |
| Parental education level (%) | | | | | | | | | | |
| Less than high school | 4% | | 25% | | 2% | | 1% | | 1% | |
| High school graduate | 32% | | 20% | | 19% | | 7% | | 4% | |
| Trade school/some college | 37% | | 30% | | 30% | | 7% | | 5% | |
| College/university degree | 16% | | 19% | | 28% | | 32% | | 31% | |
| Graduate/professional degree | 11% | | 6% | | 21% | | 53% | | 59% | |

nonsmokers along with about 11%-17% smokers in sample dataset 3, 7%-9% smokers in sample dataset 4, and 11%-15% smokers in sample dataset 5.

Table 1 also reports the descriptive statistics of covariates, including age, gender, race, and parental education level as an indicator of socioeconomic status (SES). The participants from the first three sample datasets had an average age of 15.53, 15.95, and 14.19, respectively, fitting the definition of adolescents aged between 12 and 17; and that from the sample datasets 4 and 5 had an average age of 18.28 and 18.01, respectively, fitting the definition of young adults aged between 18 and 24. Since few students in sample dataset 1 were non-white, race is not controlled in the SEM analysis and SAB models for this dataset. The parents of the participants in sample datasets 4 and 5 had higher education levels.

## Network statistics

The social networks in the five sample datasets are constructed as directed graphs. The network statistics at each wave are shown in Table 2.

The network graphs of the five sample datasets are plotted using Fruchterman-Reingold algorithm [48] in Fig 2, with circles of colors red, light green, cyan, and violet indicating increasing smoking frequency and color white indicating missing values. In Fig 2A, both non-smokers (i.e., red circles) and smokers (i.e., circles with colors other than red) in sample

**Table 2. Network statistics from the five sample datasets.**

| Sample dataset | 1 | | 2 | | 3 | | 4 | | 5 | |
|---|---|---|---|---|---|---|---|---|---|---|
| | Wave 1 | Wave 2 | Wave1 | Wave 2 | Wave 1 | Wave 2 | Wave 1 | Wave 2 | Wave 1 | Wave 2 |
| # individuals | 631 | | 1,166 | | 125 | | 196 | | 483 | |
| # ties | 2,227 | 1,964 | 2,074 | 1,697 | 253 | 332 | 796 | 674 | 3,577 | 2,990 |
| Density[a] | .006 | .005 | .002 | .001 | .016 | .021 | .021 | .018 | .015 | .013 |
| Jaccard index[b] | .25 | | .22 | | .27 | | .41 | | .34 | |
| Outdegree centrality | | | | | | | | | | |
| • Range | 0–10 | 0–10 | 0–10 | 0–10 | 0–7 | 0–8 | 0–15 | 0–13 | 0–29 | 0–25 |
| • Mean | 3.53 | 3.11 | 1.78 | 1.46 | 2.02 | 2.66 | 4.06 | 3.44 | 7.41 | 6.19 |
| • Standard deviation | 2.33 | 2.25 | 1.84 | 1.63 | 1.89 | 1.95 | 3.19 | 2.85 | 6.46 | 6.14 |
| Indegree centrality | | | | | | | | | | |
| • Range | 0–17 | 0–15 | 0–15 | 0–62 | 0–8 | 0–11 | 0–16 | 0–13 | 0–28 | 0–24 |
| • Mean | 3.53 | 3.11 | 1.78 | 1.46 | 2.02 | 2.66 | 4.06 | 3.44 | 7.41 | 6.19 |
| • Standard deviation | 2.90 | 2.66 | 1.98 | 2.43 | 1.82 | 2.58 | 3.14 | 2.81 | 4.45 | 4.00 |
| Eigenvector centrality | | | | | | | | | | |
| • Range | 0-.21 | 0-.30 | 0-.36 | 0-.68 | 0-.34 | 0-.37 | 0-.33 | 0-.36 | 0-.16 | 0-.16 |
| • Mean | .02 | .02 | .01 | .01 | .05 | .05 | .04 | .04 | .04 | .03 |
| • Standard deviation | .03 | .04 | .03 | .03 | .07 | .08 | .06 | .06 | .03 | .03 |
| Betweenness centrality | | | | | | | | | | |
| • Range | 0–18270 | 0–15825 | 0–82418 | 0–348763 | 0–3231 | 0–1739 | 0–2978 | 0–3497 | 0–8692 | 0–13808 |
| • Mean | 1967.66 | 1970.55 | 4244.64 | 3224.23 | 303.98 | 268.22 | 433.85 | 428.37 | 966.70 | 1042.99 |
| • Standard deviation | 2237.27 | 2336.77 | 7897.78 | 11582.51 | 540.91 | 375.50 | 589.70 | 635.80 | 1083.24 | 1334.07 |
| *K*-core value | | | | | | | | | | |
| • Range | 0–12 | 0–10 | 0–10 | 0–8 | 0–6 | 0–10 | 0–10 | 0–8 | 0–16 | 0–16 |
| • Mean | 6.62 | 5.66 | 3.66 | 3.01 | 4.10 | 5.31 | 5.24 | 4.50 | 11.34 | 9.56 |
| • Standard deviation | 2.62 | 2.63 | 2.32 | 2.15 | 2.11 | 2.85 | 2.78 | 2.47 | 3.82 | 3.40 |

[a]Density is the quotient of observed ties over all possible ties among individuals.

[b]Jaccard index is the quotient of persisting ties across two waves over all ties of two waves.

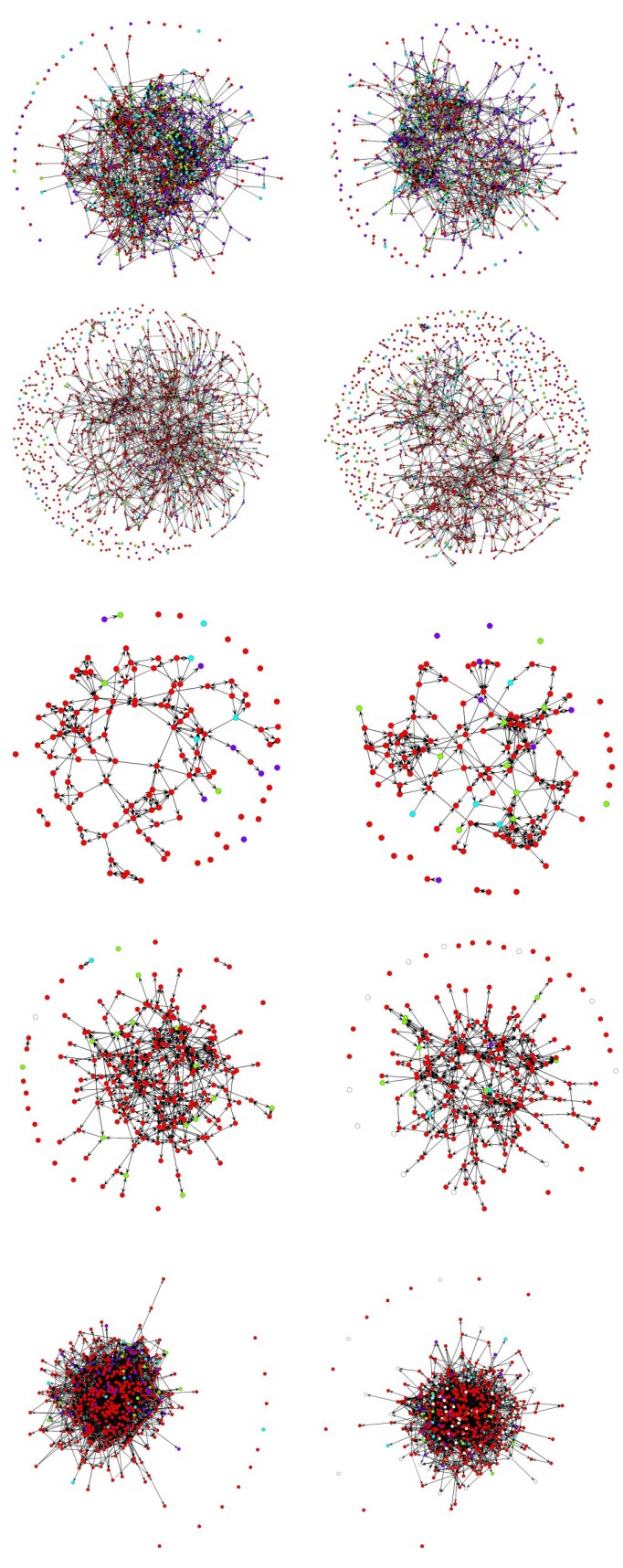

**Fig 2. Network graphs from the five sample datasets.** In network graphs red circles represent nonsmokers, light green circles represent participants who smoked 1 to 3 days in the past 30 days, cyan circles represent participants who smoked 4 to 21 days in the past 30 years, violet circles represent participants who smoked 22 or more days in the past 30 days, and white circles represent participants who had missing values.

dataset 1 seem to have an equal chance to occupy core positions in their networks, but there are more smokers showing up at margin than nonsmokers. In Fig 2B, smokers and nonsmokers in sample dataset 2 are observed at both central and peripheral positions in their networks, but nonsmokers seem to be embedded in larger $k$-cores than smokers. The pattern is not obvious among participants from the other three sample datasets by eyeballing Fig 2C–2E.

## Hypotheses I & II: SEM results

To test the first and second hypothesis, the SEM results from the five sample datasets are presented in Fig 3. For each of the 5 models, RMSEA is smaller than .06 and CFI is greater than .95, both of which suggest a good fit of the data.

When smoking was more prevalent as in sample datasets 1 and 2, smoking students at wave 1 tended to be embedded in smaller $k$-cores than nonsmokers at wave 2, as shown in Fig 3A and 3B. Therefore, Hypothesis I is not supported. Adolescent and young adult smokers stayed less central positions of their social networks, even in sample dataset 1 where over half students were smokers.

Smokers were less frequent in sample datasets 3–5. However, Hypothesis II does not receive much support. There is no disparity between smokers and nonsmokers with regards to all five inclusive network properties. Smokers and nonsmokers dispersed at random positions of their social networks when smoking was less prevalent.

Participants in sample dataset 1 with higher eigenvector centrality at wave 1 tended to have higher smoking frequency at wave 2, as shown in Fig 3A. This effect is not found in the other four sample datasets.

The SEM analyses also indicate that the covariances between the five network measures are statistically significant at wave 1, corroborating the finding of moderate to strong correlation among centrality measures in Valente et al. [49].

## Hypothesis III: Findings from SAB models

Table 3 reported the estimates from the SAB models that can be used to test Hypothesis III. The peer influence measured as an average similarity effect in the behavior equation is statistically significant in sample datasets 1 and 2, the result of which matches previous studies of these two schools [22–24, 50]. Therefore, even smokers stayed more marginal network positions than nonsmokers in sample datasets 1 and 2, they were still able to exert influence on their peers, as expected by the minority influence theory. However, no peer influence effect on smoking behavior is detected in the other three cohorts, where smokers and nonsmokers were scattered at random positions in the social networks. The sample dataset 3 was also collected during mid-1990s. Consequently, Zeitgeist is not a sufficient condition to determine the network positions of smokers and nonsmokers as well as their influence. The influence of smokers did diminish when their size decreases, not because they moved to periphery of social networks, but because the network positions of smokers and nonsmokers were now randomly mixed.

Regarding the rest of the effects in the behavior equation, the rate parameters are specified as an exponential distribution of waiting times in SAB models, suggesting that the estimated numbers of opportunities for change in smoking levels (i.e., -1 unit, no change, +1 unit) per

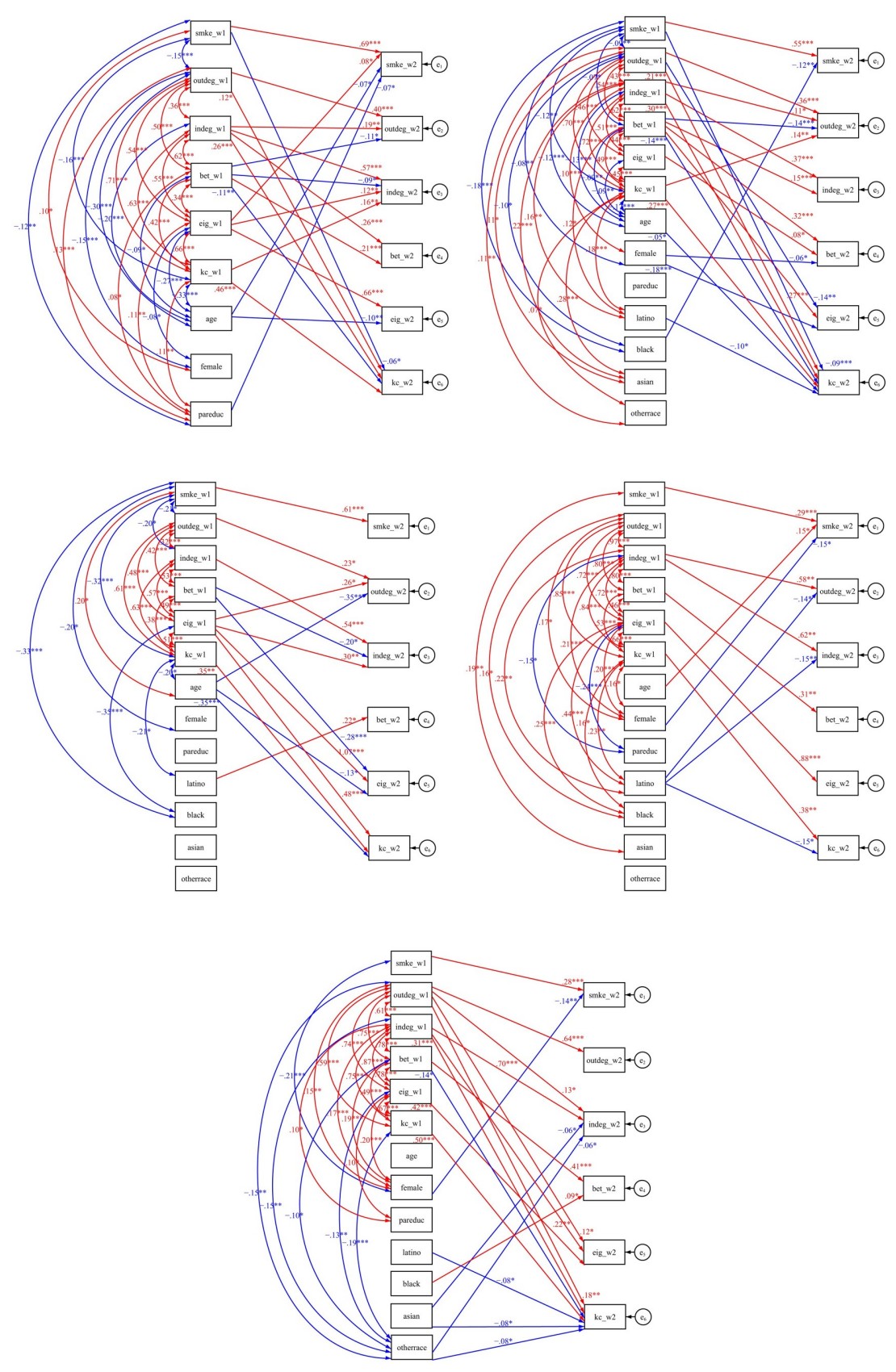

**Fig 3. SEM results from the five sample datasets.** Only statistically significant path estimates are presented. Red lines indicate positive path estimates and blue lines indicate negative path estimates. "smke"–smoking frequency; "outdeg"–outdegree centrality; "indeg"–indegree centrality; "bet"–betweenness centrality; "eig"–eigenvector centrality; "kc"–*k*-core value; "pareduc"–parental education level; "w1"–wave 1; "w2"–wave 2.

adolescent or young adult between two consecutive time points are about 14, 17, 15, 1, and 27 in the five cohorts, respectively. The rate parameter of 1 for sample dataset 4 indicates that the participants in this cohort were less likely to change their smoking levels than those in the other four cohorts. Participants in sample dataset 3 were likely to smoke less over time, as indicated by the negative linear shape parameter. Nonsmokers tended to stay as nonsmokers and smokers tended to stay as smokers, as indicated by the positive quadratic shape parameters in most cohorts. Participants in sample dataset 5 with higher indegree centrality values and who were males had higher smoking levels than their counterparts. Black students in sample dataset 2 smoked less than white students.

In the network equation, the rate parameters suggest that the estimated numbers of opportunities for change in network ties (i.e., -1 tie, no change, +1 tie) per adolescent or young adult between two observation time points are about 15, 10, 9, 11, and 26 in the five cohorts,

**Table 3. Results from Stochastic Actor-Based models.**

| Sample dataset | 1 | 2 | 3 | 4 | 5 |
|---|---|---|---|---|---|
| Behavior equation | beta (s.e.) | beta (s.e.) | beta (s.e.) | beta (s.e.) | beta (s.e.) |
| Rate parameter | 13.72*** (1.32) | 17.18*** (4.10) | 14.93*** (2.87) | 1.44*** (0.42) | 27.07*** (4.21) |
| Linear shape | -0.50 (0.34) | -1.74 (1.21) | -3.59** (1.23) | -7.83 (5.56) | -5.30 (5.42) |
| Quadratic shape | 0.65*** (0.03) | 0.70*** (0.05) | 0.91*** (0.11) | 2.22† (1.23) | 1.32*** (0.10) |
| Average similarity | 1.94***(0.29) | 1.56* (0.68) | -0.11 (1.52) | 3.19 (3.54) | -15.10 (9.16) |
| Indegree | 0.01 (0.01) | 0.01 (0.03) | -0.03 (0.06) | 0.14 (0.22) | 0.11* (0.04) |
| Female (Yes = 1) | 0.01 (0.05) | -0.01 (0.06) | -0.23 (0.21) | -7.03 (5.72) | -1.45*** (0.39) |
| Age | 0.01 (0.02) | 0.02 (0.06) | 0.11† (0.06) | 0.17 (0.72) | -0.21 (0.26) |
| Parental education | -0.02(0.04) | -0.01 (0.04) | -0.05 (0.12) | 0.06 (0.40) | 0.03 (0.25) |
| Black (Yes = 1) | | -0.28*** (0.08) | -0.44 (0.32) | 1.95 (2.16) | 0.24 (0.52) |
| Latino (Yes = 1) | | -0.02 (0.07) | -0.06 (0.34) | 0.54 (1.53) | 0.28 (0.35) |
| Network equation | beta (s.e.) | beta (s.e.) | beta (s.e.) | beta (s.e.) | beta (s.e.) |
| Rate parameter | 14.77*** (0.55) | 10.20*** (1.98) | 9.23*** (0.84) | 11.32*** (0.86) | 25.62*** (1.02) |
| Outdegree (density) | -3.94*** (0.09) | -5.10*** (0.19) | -3.40*** (0.13) | -5.62*** (0.28) | -3.14*** (0.04) |
| Reciprocity | 3.01*** (0.10) | 3.06*** (0.40) | 2.21*** (0.22) | 6.84*** (0.38) | 2.98*** (0.08) |
| Transitive triplets | 0.92*** (0.04) | 1.27*** (0.06) | 0.86*** (0.08) | 0.54 (0.48) | 0.67*** (0.04) |
| Transitive reciprocated triplets | -1.18*** (0.07) | -1.33** (0.47) | -1.10*** (0.18) | -0.84*** (0.25) | 0.07 (0.08) |
| 3-cycles | 0.31*** (0.09) | 0.03 (0.58) | 0.24 (0.21) | 0.76 (0.92) | -1.05*** (0.17) |
| Age similarity | 2.89*** (0.21) | 2.30*** (0.25) | 2.26*** (0.40) | 0.11 (0.20) | -0.02 (0.11) |
| Gender similarity | 0.28*** (0.04) | 0.37*** (0.06) | 0.05 (0.10) | 0.20* (0.10) | 0.10* (0.05) |
| Same race | | 1.05*** (0.08) | 0.63*** (0.13) | 0.21* (0.10) | 0.08† (0.04) |
| Parental education similarity | 0.23** (0.07) | 0.08 (0.19) | 0.48* (0.19) | -0.09 (0.24) | -0.08 (0.11) |
| Smoking behavior alter | 0.18** (0.06) | 0.02 (0.11) | 0.21 (0.16) | 0.63 (0.50) | -0.16 (0.20) |
| Smoking behavior ego | -0.10† (0.06) | 0.09 (0.14) | -0.19 (0.17) | 1.89* (0.95) | -0.08 (0.21) |
| Smoking behavior similarity | 1.62*** (0.28) | 1.41** (0.51) | 0.46 (0.47) | 3.62 (2.23) | -0.46 (0.66) |

† Two-sided p<0.1

* Two-sided p<0.05

** Two-sided p<0.01

*** Two-sided p<0.001.

**Table 4. Results from Stochastic Actor-Based models decomposing increase (creation function) and decrease (endowment function).**

| Sample dataset | 1 | 2 |
|---|---|---|
| Behavior equation | beta (s.e.) | beta (s.e.) |
| Average similarity (creation) | 7.72* (3.45) | -10.14* (5.02) |
| Average similarity (endowment) | 2.06* (1.03) | 7.63* (3.70) |

* Two-sided p<0.05.

respectively. They didn't select random peers as friends but preferred mutual interactions, as indicated by the negative outdegree (density) parameters and positive reciprocity parameters. After the transitive reciprocated triplets effect is controlled as suggested by Block [51], adolescents and young adults tended to select a friend's friend as a friend, as indicated by the positive transitive triplets parameters in most cohorts. The homophilous selection effect based on age is evident in three cohorts, that on gender is evident in four cohorts, that on race is evident in three cohorts, and that on parental education level is evident in two cohorts. Smokers were more popular in sample dataset 1, as indicated by the positive smoking behavior alter parameter. Smokers tended to form and maintain more friendship ties in sample dataset 4, as indicated by the positive smoking behavior ego effect. Finally, homophilous selection effect based on smoking behavior is statistically significant in sample datasets 1 and 2.

To evaluate the magnitudes of peer influence from smokers and nonsmokers, we tweak the SAB models for sample datasets 1 and 2 as shown in Table 3 by decomposing the create function and the endowment function, with the former indicating to what extent smokers influence their friends to increase smoking frequency (+1 unit vs. no change) and the latter indicating to what extent nonsmokers influence their friends to decrease smoking frequency (-1 unit vs. no change). As shown in Table 4, students in sample datasets 1 and 2 were more likely to be affected by their peers to start smoking than to quit smoking, which corroborates the finding in Hass and Schaefer [11]. Taking together, while nonsmokers influenced their friends to stop smoking or decrease smoking frequency, smokers also influenced their friends to start smoking or increase smoking frequency. The magnitude of the latter influence could be even greater, despite the relatively peripheral positions of smokers in their social networks. This finding is against the main statement of Hypotheses III.

Before discussing the findings of this study, Table 5 shows six possible scenarios combing the smoking prevalence (high vs. low) and network positions of smokers (core, periphery, vs. random) and highlights the common patterns of the five sample datasets: Sample datasets 1 and 2 fall in the cell of high smoking prevalence and peripheral network positions, and sample datasets 3 to 5 fall in the cell of low smoking prevalence and random network positions. No sample dataset falls into the other four cells in Table 5.

## Discussion

We expected to see adolescent and young adult smokers to be at core of social networks as their adult counterparts [2] when smoking was prevalent, e.g., in sample datasets 1 and 2. By

**Table 5. Crosstab of smoking prevalence and network positions of smokers.**

| Network position<br><br>Smoking prevalence | Core | Periphery | Random |
|---|---|---|---|
| High | | Sample datasets 1–2 | |
| Low | | | Sample datasets 3–5 |

the same token, we supposed smokers to be at periphery of social networks and thus less influential when there were fewer of them, e.g., in sample datasets 3–5. However, none of these hypotheses was supported by findings from the five sample datasets we constructed.

Instead, our findings from sample datasets 1 and 2 indicated that adolescent and young adult smokers tended to be embedded in smaller $k$-cores than nonsmokers when smoking was relatively prevalent. Moreover, smokers were able to exert influence on their peers, just like nonsmokers embedded in larger $k$-cores, and the magnitude of influence from smokers could be larger than that from nonsmokers [11] in spite of the smokers' marginal positions in social networks. When smoking was less prevalent in sample datasets 3–5, smokers and nonsmokers connected with one another with no significant difference in network positions, and thus peer influence was not operating on smoking behavior via social networks.

Therefore, peer influence on smoking behavior could work differently for adolescents/ young adults and adults. Baker and Faulkner [52] have found that individuals engaging in illegal behaviors intentionally created sparse and decentralized networks to avoid detection, prosecution, and sanctioning. Similarly, to conceal their smoking activities from school administrators, teachers, and staffs, adolescent and young adult smokers might want to act as peripheral players, occupied decentralized network positions, and formed relatively isolated sub-communities. In this way, they could reduce their vulnerability and risk of exposure, work together against school disciplines, maintain their behavior and local norm, exchange mutual support, and exert peer influence. Instead, adult smokers faced no such punishment and stress and thus did not have to hide their smoking behavior from anyone as in Christakis and Fowler [2].

Although network positions of adolescent and young adult smokers and nonsmokers did not work the way as we have hypothesized, they opened and inspected what has been treated as a "black box" in minority influence theory and peer influence literature. It does not matter much to what extent the adolescent and young adult smokers are majority or minority in the population. It is all about the smokers' and nonsmokers' positions in the social networks. Both smokers and nonsmokers have to occupy distinct network positions to be influential. Otherwise, if smokers and nonsmokers are dispersed at random positions of social networks, which is more likely to happen when smoking is less frequent, neither of them can exert a remarkable degree of peer influence.

We anticipate our findings can be examined across a wider range of longitudinal datasets collected on US adolescents and young adults, especially nationally representative ones. There are at least two other related case studies. Mathys et al. [53] studied a longitudinal sample of 450 students moving from 10th grade to 11th grade of a public high school in a US mid-sized Northeastern city between 2004 and 2005. The percentages of smokers were about 16% and 19% at wave 1 and wave 2, respectively, and no peer influence effect on smoking behavior was detected in the sample. The PROSPER study [54, 55] starting from 2006 is a prevention trial in 28 small public-school districts, half in Iowa and half in Pennsylvania. The initial past-month cigarette use rate was 7%-8% [54]. But it kept increasing from wave 1 to wave 6, and the peer influence effect on smoking has been reported by Osgood et al. [55]. We have no access to these two datasets. Future research is needed to examine the findings from the current study in more robust and generalizable settings.

Two limitations of this study warrant consideration. First, the networks from the Add Health study were constructed from surveys of high school students, while that in the NetSense and NetHealth projects were generated from communication events between undergraduate students in an elite private university. The differences in developmental period (adolescence to young adulthood vs. young adulthood), context (high school vs. college), and method (obtrusive vs. unobtrusive measurement) could yield some degree of bias to our findings. Future

work should aim at collecting longitudinal network and smoking behavior data free of such potential bias. Second, a multilevel or fixed-effect modeling framework would be more appropriate to test our hypotheses, but we only have accessibility to five datasets and thus the puzzle is not complete yet. While this study might disclose part of the black box, fully understanding of the evolutionary patterns of smoking prevalence and network positions needs collaboration of scientists in a wide range of disciplines.

With regard to preventive implications, Valente [56] has argued that interventions leveraged with social networks for health behavior changes may be superior to non-network-based interventions. Our findings suggested that when smokers exert stronger peer influence than nonsmokers, e.g., in sample datasets 1 and 2, it is important to suppress the influence from the former and strengthen the influence from the latter. However, when peer influence mechanism stops working, e.g., in sample datasets 3–5, more individual-targeted intervention strategies should be taken into consideration by prevention scientists.

## Supporting information

**S1 File. Data for the NetSense study participants (netsense.Rdata).**
(ZIP)

## Author Contributions

**Conceptualization:** Cheng Wang.

**Data curation:** Cheng Wang.

**Formal analysis:** Cheng Wang.

**Methodology:** Cheng Wang.

**Software:** Cheng Wang.

**Validation:** Cheng Wang.

**Visualization:** Cheng Wang.

**Writing – original draft:** Cheng Wang.

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
