## [Decision Letter · Decision Letter 0]

18 Nov 2020

PONE-D-20-32056

Smoking prevalence, core/periphery network positions, and peer influence: findings from five datasets on US adolescents

PLOS ONE

Dear Dr. Wang,

Thank you for submitting your manuscript to PLOS ONE. After careful consideration, we feel that it has merit but does not fully meet PLOS ONE’s publication criteria as it currently stands. Therefore, we invite you to submit a revised version of the manuscript that addresses the points raised during the review process.

Note that in addition to comments entered into Editorial Manager Reviewer 1 provided a Word file with more comments.

We look forward to receiving your revised manuscript.

Kind regards,

Stanton A. Glantz

Academic Editor

PLOS ONE

Journal Requirements:

Reviewers' comments:

Reviewer's Responses to Questions

**Comments to the Author**

1. Is the manuscript technically sound, and do the data support the conclusions?

Reviewer #1: Partly

Reviewer #2: Partly

2. Has the statistical analysis been performed appropriately and rigorously? 

Reviewer #1: Yes

Reviewer #2: Yes

3. Have the authors made all data underlying the findings in their manuscript fully available?

Reviewer #1: Yes

Reviewer #2: Yes

4. Is the manuscript presented in an intelligible fashion and written in standard English?

Reviewer #1: Yes

Reviewer #2: Yes

5. Review Comments to the Author

Reviewer #1: 1. The analytical techniques are technically sound. However, I do have concerns regarding the generalisability of the sample (i.e. the social networks are different types of networks, the age of the participants varies considerably, and Add Health was collected in a different time period compared to NetSense/NetHealth).

2. The statistical analysis is comprehensive and rigorous.

3. All findings are presented and described. As stated in my comments to the author, I believe that the results could be more concise.

4. As partly was not an option, I selected yes. However, there is a need for careful spellcheck of the English language throughout. There are also multiple grammatical errors (multiple sentences begin with 'And').

In addition, my comments to the author are uploaded as an attachment.

Reviewer #2: This is a very interest and relevant paper designed to understand how smoking works in adolescent networks. Though the data is somewhat dated, describing an underlying interpersonal process is important.

Though the literature review is not exhaustive, it touches on the most significant areas that need to be addressed.

In the introduction and abstract the author writes as if the paper is about adolescents, but it is actually about adolescents and young adults. Therefore, both literatures should be incorporated.

There are some points of clarification that would help in the methods. First, the authors clearly elucidate a data collection issue that they are avoiding based on a cutoff. Three schools were chosen. How many schools met the 85% cutoff? What additional criteria were applied to select amongst those or were there only 3 available schools?

In the methods you introduce a variety of analyses and strategies. Each of these should include a clear statement about why it was chosen, where it has been used before and which hypothesis it is linked to. This will help later as you go through the very long results section.

Then, when you get into the results, it might be helpful to organize them by hypotheses explicitly instead of approach. I understand why you have it organized the way you do, but there’s a lot going on and it’s easy to get lost in the weeds of the results section.

A note about the results and discussion, you switch between calling the Notre Dame date University of Notre or naming it by the two dataset contained therein. Pick one and stick with it.

Page 17, in the T-test results section, there are a number of sentences with words missing (e.g. Regarding the rest network statistics,)

On page 12, the following sentence should be rewritten from, “The peer influence measured as average” to “The peer influence was measured as an average”

On page 12, the following sentence is missing a awkward, “the result of which matches that in previous studies of these two schools.” It should be reworded as, “the result of which matches previous studies of these two schools.”

One page 13, the following phrase, “Regarding the rest effects in the behavior equation” should read, “Regarding the rest of the effects in the behavior equation”

Also on page 13 where you are talking about the SAB models. It would be helpful to have some explanation of how to interpret the following, “Regarding the rest effects in the behavior equation, the rate parameters suggest that adolescents had about 14, 17, 15, 1, and 27 opportunities to change their smoking levels in the five cohorts, respectively.” I am not terribly familiar with these models and I suspect many of your readers will not be as well. What does the rate parameter tell us? Is the difference between 1 and 27 substantial or not?

On page 14, you say that you did something by “decomposing the create function and the endowment function” You define these in the methods, but after seeing them applied, you might want to remind readers what they.

Why did you do the analysis for Table 6? What does this add to the paper and how does it help you test your hypotheses?

The discussion seems appropriate. In the last paragraph, however, the author makes a statement about policy implication. Designing an intervention to target particular individuals is not policy, it’s an intervention and should called thus.

6. PLOS authors have the option to publish the peer review history of their article (what does this mean?). If published, this will include your full peer review and any attached files.

Reviewer #1: No

Reviewer #2: No

---

## [Author Response · Author response to Decision Letter 0]

2 Jan 2021

Response to Reviewers

PONE-D-20-32056

Smoking prevalence, core/periphery network positions, and peer influence: findings from five datasets on US adolescents

PLOS ONE

I would like to thank the editor and reviewers for their very constructive feedback. Incorporating the reviewers’ suggestions for revision has resulted in a greatly improved manuscript. Below, I note the concerns of the reviewers and then explain how I responded to each comment. 

Reviewer 1

1. Careful spellcheck required, alongside a grammar check (several sentences begin with ‘And’). 

––––––––––––––––––––––––––––––––––––––––––––––––––––

I thank the reviewer for this suggestion. I have conducted a thorough spellcheck and grammar check, with all sentences that begin with “And” revised.

2. Requires consistency in the reference to schools – you both name them individually or refer to them as sample datasets 1-5.

––––––––––––––––––––––––––––––––––––––––––––––––––––

This is a valid point. In the revised manuscript I consistently refer to them as sample datasets 1-5.

3. Page 4: Social networks should be introduced prior to outlining the hypotheses. You assume that the reader has prior knowledge of social network terminology, but do not define or explain social network concepts (i.e. network positions and peer influence processes).

––––––––––––––––––––––––––––––––––––––––––––––––––––

This is a critical comment. Members occupying core positions form a central and densely connected cluster in a social network, and those occupying peripheral positions of a network have just a few links or no connections at all. I now explain these two network position concepts in the second paragraph on page 3 in the revised manuscript.

Peer influence refers to the propensity of a person to alter his or her behavior or attitude such that it becomes more similar to that of his or her contacts. I now explain this concept in the last paragraph on page 4 in the revised manuscript.

4. Page 5: The paragraph outlining minority influence is confusing – can you explain what you mean by ‘it is assumed the majority and minority can both be influential’. The terms ‘minority’ and ‘majority’ should be defined.

––––––––––––––––––––––––––––––––––––––––––––––––––––

Majority is the greater (more than half) subset of individuals and minority is the smaller (less than half) subset of individuals in a population. I now clarify these two terms in the second paragraph on page 5 in the revised manuscript.

5. Page 6: consider re-phrasing the term ‘disclose the black box in previous research’.

––––––––––––––––––––––––––––––––––––––––––––––––––––

I revise this sentence as “in this study we aim to extend this line of research with several empirical datasets” on page 6.

6. Page 6: ‘The students in participating schools were asked to report their smoking frequency in the past 30 days.’ – this information is repeated on page 8 ‘Smoking Measures’.

––––––––––––––––––––––––––––––––––––––––––––––––––––

I remove the duplicated part and revise this sentence as “The students in participating schools had an average smoking prevalence of 25%” on page 6.

7. Page 8: You do not provide any information on the social network questions included in the NetSense and NetHealth datasets. You cannot make comparisons across different types of networks, only if they too are friendship networks (to be consistent with Add Health networks).

––––––––––––––––––––––––––––––––––––––––––––––––––––

I am sorry for the confusion. The reviewer is right that NetSense and NetHealth project used a different strategy from the Add Health study to collect information on friendship networks. The Add Health study used the “name generator”, i.e., a question for the participants to nominate up to five female and five male friends, as indicated on page 7. Instead, the NetSense and NetHealth projects built friendship networks among their participants with real-word communication events that were aggregated from the call detailed records (CDRs) retrieved from the participants’ smartphones, as indicated on page 8.

8. Page 9: Analytical strategy – given the demographic variance in your datasets, why have you not controlled for characteristics such as parental education and race?

––––––––––––––––––––––––––––––––––––––––––––––––––––

I now clarify on page 10 that “Given the demographic variance in the five sample dataset, we also control for covariates such as gender, age, parental education, and race in the SAB models”. 

9. Page 11: the results section is generally hard to follow as you report a lot of results but do not explain them in detail. What relevance is there in reporting the cross-sectional and t-test results? It is difficult to follow all of the results as they are erratic and confusing – you report results from some schools and not others. In general, it is my recommendation that the results section could be refined to be more concise.

––––––––––––––––––––––––––––––––––––––––––––––––––––

I follow the reviewer’s suggestion and remove the t-test part from the “Results” section in the revised manuscript.

10. Page 13: ‘Regarding the rest effects in the behaviour equation’ – can you clarify what this means? 

––––––––––––––––––––––––––––––––––––––––––––––––––––

I revise it to “Regarding the rest of the effects in the behavior equation”.

11. Page 13: ‘in the network equation, the rate parameters suggest that adolescents had about 15, 10, 9 11 and 26 opportunities…’ – it is unclear what you mean by this statement.

––––––––––––––––––––––––––––––––––––––––––––––––––––

I am sorry for the confusion. I now revise it as “In the network equation, the rate parameters suggest that the estimated numbers of opportunities for change in network ties (i.e., -1 tie, no change, +1 tie) per adolescent between two observation time points are about 15, 10, 9, 11, and 26 in the five cohorts, respectively”. 

12. Page 16-17: ‘we have no access to these two …but there is a good chance’ – you should not make inferences in this case.

––––––––––––––––––––––––––––––––––––––––––––––––––––

I revise it to “We have no access to these two datasets. Future research is needed to examine the findings from the current study in more robust and generalizable settings.”

13. Page 17: Only in the discussion do you address the disparity of the social network questions and the age range of participants. The Add Health study measures a completely different network to the NetSense and NetHealth studies – this is highly important and raises the question of how representative the results are when you have two different networks. Furthermore, whilst you address the difference in age (Add Health participants are young adolescents compared to late adolescents/emerging adults in NetSense and NetHealth) you do not address the time frame. It is important to consider the implications of the different periods in time that the studies were set.

––––––––––––––––––––––––––––––––––––––––––––––––––––

I totally agree with the reviewer there are various types of disparities across the Add Health, NetSense, and NetHealth studies. In the manuscript I honestly discussed their differences in how social networks were constructed, age composition of participants, and time frame of each sample dataset. I also controlled the demographic variance in the five sample dataset, including covariates such as gender, age, parental education, and race in the SEM analyses and SAB models.

Having said that, we lack nationally representative samples continually covering US adolescents in the past 30 years that allow us to investigate the evolutionary patterns of smoking prevalence, network positions, and peer influence with aforementioned disparities fully considered. The Add Health data were collected in 1995 and 1996, the NetSense data were collected in 2011 and 2012, and the NetHealth data were collected in 2015 and 2016. In this study there is no dataset from the first decade of the 21st century. I have contacted the authors of Mathys et al. (2013) but got no response. While PROSPER data are available for researchers at Penn State University, limited accessibility is provided for scholars outside Penn State. Due to the constraint in data availability, I am very cautious and not claiming that I am examining the changing patterns of US adolescents from different periods in time. Therefore, the title of this manuscript is “Smoking prevalence, core/periphery network positions, and peer influence: findings from five datasets on US adolescents”. 

To sum up, this study aims to present some common patterns of the five sample datasets, with various types of disparities acknowledged. I expect these patterns to be confirmed or challenged by collaboration of scientists across a wide range of disciplines in the near future.

Reviewer 2

1. In the introduction and abstract the author writes as if the paper is about adolescents, but it is actually about adolescents and young adults. Therefore, both literatures should be incorporated.

––––––––––––––––––––––––––––––––––––––––––––––––––––

We thank the reviewer for this suggestion. As shown in Table 1, the participants had a range of age between 14-18 in sample dataset 1, 12-19 in sample dataset 2, 11-18 in sample dataset 3, 17-19 in sample dataset 4, and 17-20 in sample dataset 5.

The US Food and Drug Administration (FDA) defines adolescents as individuals in the 12-21 years age group (FDA 2003). American Academy of Pediatrics (AAP) identifies adolescence as 11 to 21 years of age, including early (ages 11-14 years), middle (ages 15-17 years), and late (ages 18-21years) adolescence (Hagan et al. 2008).

Therefore, I consider the participants in the five sample datasets as adolescents. But I clarify the FDA and AAP’s definitions of adolescents on pages 10-11 in the revised manuscript when introducing the age composition of the participants in Table 1.

2. There are some points of clarification that would help in the methods. First, the authors clearly elucidate a data collection issue that they are avoiding based on a cutoff. Three schools were chosen. How many schools met the 85% cutoff? What additional criteria were applied to select amongst those or were there only 3 available schools?

––––––––––––––––––––––––––––––––––––––––––––––––––––

This is a great question. After excluding school 175, only three out of fifteen saturated schools – Jefferson High, Sunshine High, and school 28 – met the 85% cutoff. 

As we introduced in the manuscript, the Add Healthy study suffered from an administrative error, i.e., some students were only allowed to ask one female and one male best friends during the wave 1 In-Home survey. While missing data is a well-known challenge for traditional cross-sectional and longitudinal studies, the problem is even greater for network studies. As indicated in Handcock (2002), if we lack information on f participants in a network, the fraction of unobserved edges will be 1-(1-f)^2. The following table shows four schools with the lowest proportion of limited nomination. In school 28, about 14% adolescents encountered the limited nomination issues and 27% edges were unobserved; but in school 88, about 26% adolescents encountered the limited nomination issue and 45% edges were unobserved. Therefore, we set cutoff at 85% and only pick three schools from the Add Healthy study.

Add Health School % limited nomination (f) % unobserved edges (1-(1-f)^2) 

Sunshine High 5.23% 10.19% 

Jefferson High 5.71% 11.09% 

28 14.40% 26.72% 

88 26.03% 45.28% 

3. In the methods you introduce a variety of analyses and strategies. Each of these should include a clear statement about why it was chosen, where it has been used before and which hypothesis it is linked to. This will help later as you go through the very long results section.

––––––––––––––––––––––––––––––––––––––––––––––––––––

This is a crucial comment. To test the first and second hypotheses, I estimate a structural equation model (SEM) for each sample dataset using Stata V16.0 because we have multiple causes and multiple outcomes simultaneously. The SEM analysis is appropriate for longitudinal study of the multiple-path relationships between adolescent peer networks and their smoking behavior (Hall & Valente 2007; Lakon & Hipp 2014).

To test the third hypothesis, I adopt the Stochastic Actor-Based (SAB) modeling strategy implemented in RSiena. The SAB modeling strategy has been widely used to investigate the peer influence via social networks on adolescent smoking behavior in recent years (Shoham et al. 2012; Schaefer et al. 2012; Mathys et al. 2013; Haas & Schaefer 2014; Lakon et al. 2014; Osgood et al. 2015; Wang et al. 2016, 2018; Montgomery et al. 2020). The average similarity effect indicates whether peer influence exists in an environment or not. The creation function is used to measure the magnitude of peer influence on increasing smoking levels and the endowment function is used to measure the magnitude of peer influence on decreasing smoking levels (Haas & Schaefer 2014). 

I now clarify these details under the subheader “Analytical Strategy” in “Materials and methods” section on pages 9-10.

4. Then, when you get into the results, it might be helpful to organize them by hypotheses explicitly instead of approach. I understand why you have it organized the way you do, but there’s a lot going on and it’s easy to get lost in the weeds of the results section.

––––––––––––––––––––––––––––––––––––––––––––––––––––

I now follow the reviewer’s suggestion and revise the subheaders to “Hypotheses I & II: SEM Results” on page 11 and “Hypothesis III: Findings from SAB Models” on page 12 in the “Results” section.

5. A note about the results and discussion, you switch between calling the Notre Dame date University of Notre or naming it by the two dataset contained therein. Pick one and stick with it.

––––––––––––––––––––––––––––––––––––––––––––––––––––

As the reviewer has suggested, I stick to sample datasets 4 and 5 in the “Results” and “Discussion” section in the revised manuscript.

6. Page 17, in the T-test results section, there are a number of sentences with words missing (e.g. Regarding the rest network statistics). 

––––––––––––––––––––––––––––––––––––––––––––––––––––

I follow the suggestion from reviewer 1 and remove the t-test part from the “Results” section in the revised manuscript.

7. On page 12, the following sentence should be rewritten from, “The peer influence measured as average” to “The peer influence was measured as an average”. 

––––––––––––––––––––––––––––––––––––––––––––––––––––

I revise this sentence on page 12 as the reviewer suggested.

8. On page 12, the following sentence is missing a awkward, “the result of which matches that in previous studies of these two schools.” It should be reworded as, “the result of which matches previous studies of these two schools”. 

––––––––––––––––––––––––––––––––––––––––––––––––––––

I revise this sentence on page 12 as the reviewer suggested.

9. One page 13, the following phrase, “Regarding the rest effects in the behavior equation” should read, “Regarding the rest of the effects in the behavior equation”. 

––––––––––––––––––––––––––––––––––––––––––––––––––––

I revise this sentence on page 13 as the reviewer suggested.

10. Also on page 13 where you are talking about the SAB models. It would be helpful to have some explanation of how to interpret the following, “Regarding the rest effects in the behavior equation, the rate parameters suggest that adolescents had about 14, 17, 15, 1, and 27 opportunities to change their smoking levels in the five cohorts, respectively.” I am not terribly familiar with these models and I suspect many of your readers will not be as well. What does the rate parameter tell us? Is the difference between 1 and 27 substantial or not. 

––––––––––––––––––––––––––––––––––––––––––––––––––––

I am sorry for the confusion. I now revise it as “the rate parameters are specified as an exponential distribution of waiting times in SAB models, suggesting that the estimated numbers of opportunities for change in smoking levels (i.e., -1 unit, no change, +1 unit) per adolescent between two consecutive time points are about 14, 17, 15, 1, and 27 in the five cohorts, respectively. The rate parameter of 1 for sample dataset 4 indicates that the participants in this cohort were less likely to change their smoking levels than those in the other four cohorts.” 

11. On page 14, you say that you did something by “decomposing the create function and the endowment function” You define these in the methods, but after seeing them applied, you might want to remind readers what they. 

––––––––––––––––––––––––––––––––––––––––––––––––––––

Again, a valid point. The creation function indicates to what extent smokers influence their friends to increase smoking frequency (+1 unit vs. no change), and the endowment function indicates to what extent nonsmokers influence their friends to decrease smoking frequency (-1 unit vs. no change). I now reiterate these details on page 14 in the revised manuscript.

12. Why did you do the analysis for Table 6? What does this add to the paper and how does it help you test your hypotheses? 

––––––––––––––––––––––––––––––––––––––––––––––––––––

Table 6 summarizes the common patterns in findings from the inclusive five sample datasets. As indicated in my reply to comment 13 from reviewer 1, I expect these patterns to be confirmed or challenged by collaboration of scientists across a wide range of disciplines in the near future.

13. The discussion seems appropriate. In the last paragraph, however, the author makes a statement about policy implication. Designing an intervention to target particular individuals is not policy, it’s an intervention and should called thus. 

––––––––––––––––––––––––––––––––––––––––––––––––––––

I am thankful for a careful reading of the manuscript. I now revise “policy implications” to “preventive implications” in the last paragraph of the “Discussion” section.

References

Haas SA, Schaefer DR. With a little help from my friends? Asymmetrical social influence on adolescent smoking initiation and cessation. J Health Soc Behav. 2014; 55: 126–143.

Hagan JF Jr, Shaw JS, Duncan P, editors. Bright Futures: Guidelines for the Health Supervision of Infants, Children, and Adolescents. 3rd ed. Elk Grove Village, IL: American Academy of Pediatrics; 2008.

Hall JA, Valente TW. Adolescent smoking networks: The effects of influence and selection on future smoking. Addict Behav. 2007; 32: 3054–3059.

Handcock MS. Missing Data for of Social Networks. Center for Statistics and the Social Sciences, University of Washington; 2002.

Lakon CM, Hipp JR. On social and cognitive influences: Relating adolescent networks, generalized expectancies, and adolescent smoking. PLoS ONE. 2014; 9: e115668.

Lakon CM, Wang C, Butts CT, Jose R, Timberlake DS, Hipp JR. A dynamic model of adolescent friendship networks, parental influences, and smoking. J Youth Adolesc. 2014; 44: 1767–1786.

Mathys C, Burk WJ, Cillessen AHN. Popularity as moderator of peer selection and socialization of adolescent alcohol, marijuana, and tobacco use. J Res Adolesc. 2013; 23: 513–523.

Montgomery SC, Donnelly M, Bhatnagar P, Carlin A, Kee F, Hunter RF. Peer social network processes and adolescent health behaviors: A systematic review. Prev Med. 2020; 130: 105900.

Osgood DW, Feinberg ME, Ragen DT. Social networks and the diffusion of adolescent problem behavior: Reliable estimates of selection and influence from sixth through ninth grades. Prev Sci. 2015; 16: 832–842. 

Schaefer DR, Haas SA, Bishop N. A dynamic model of US adolescents’ smoking and friendship networks. Am J Public Health. 2012; 102: e12–e18.

Shoham DA, Tong L, Lamberson PJ, Auchincloss AH, Zhang J, Dugas L, et al. An actor-based model of social network influence on adolescent body size, screen time, and playing sports. PLoS ONE. 2012; 7: e39795. 

US Food and Drug Administration (FDA). Guidance for Industry and FDA Staff: Pediatric Expertise for Advisory Panels. Rockville, MD: US Department of Health and Human Services, Food and Drug Administration, Center for Devices and Radiological Health; 2003. Available from: https://www.fda.gov/media/72451/download.

Wang C, Hipp JR, Butts CT, Jose R, Lakon CM. Coevolution of adolescent friendship networks and smoking and drinking behaviors with consideration of parental influence. Psychol Addict Behav. 2016; 30: 312–324. 

Wang C, Hipp JR, Butts CT, Lakon CM. The interdependence of cigarette, alcohol, and marijuana use in the context of school-based social networks. PLoS ONE. 2018; 13: e0200904.

---

## [Decision Letter · Decision Letter 1]

26 Jan 2021

PONE-D-20-32056R1

Smoking prevalence, core/periphery network positions, and peer influence: findings from five datasets on US adolescents

PLOS ONE

Dear Dr. Wang,

Thank you for submitting your manuscript to PLOS ONE. After careful consideration, we feel that it has merit but does not fully meet PLOS ONE’s publication criteria as it currently stands. Therefore, we invite you to submit a revised version of the manuscript that addresses the points raised during the review process.

Please pay particular attention to Reviewer 2's request to divide adolescents and young adults conceptually.

We look forward to receiving your revised manuscript.

Kind regards,

Stanton A. Glantz

Academic Editor

PLOS ONE

Reviewers' comments:

Reviewer's Responses to Questions

**Comments to the Author**

1. If the authors have adequately addressed your comments raised in a previous round of review and you feel that this manuscript is now acceptable for publication, you may indicate that here to bypass the “Comments to the Author” section, enter your conflict of interest statement in the “Confidential to Editor” section, and submit your "Accept" recommendation.

Reviewer #1: All comments have been addressed

Reviewer #2: (No Response)

2. Is the manuscript technically sound, and do the data support the conclusions?

Reviewer #1: Yes

Reviewer #2: Yes

3. Has the statistical analysis been performed appropriately and rigorously? 

Reviewer #1: Yes

Reviewer #2: Yes

4. Have the authors made all data underlying the findings in their manuscript fully available?

Reviewer #1: Yes

Reviewer #2: Yes

5. Is the manuscript presented in an intelligible fashion and written in standard English?

Reviewer #1: Yes

Reviewer #2: Yes

6. Review Comments to the Author

Reviewer #1: I commend you for your careful consideration of my suggestions and comments. I am satisfied that all of my queries have been addressed.

Reviewer #2: The paper is much improved.

A couple additional comments.

I appreciate the citations in the methods about adolescence. However, tobacco control researchers, who will be reading this if it is published, often treat individuals 18-24 as separate from adolescents. It’s fine to include older young adults in this analysis and I understand the NIH and FDA and other organizations treat them as children but your audience will not. This paper will be more credible to your audience if you engage with the norms they use the talk about this population. It would not be hard to reframe the literature review. This would also give you the opportunity to incorporate, more broadly, relevant literature on smoking behavior in your population of interest. You could incorporate this literature and then say that you are going to use the term adolescent but this young adult population is of specific interest in tobacco control and other areas. Please see the following citations:

Smoking impacts on prefrontal attentional network function in young adult brains, Francesco Musso, Franziska Bettermann, Goran Vucurevic, Peter Stoeter, Andreas Konrad & Georg Winterer, Psychopharmacology volume 191, pages159–169(2007

American Journal of Preventive Medicine, Volume 53, Issue 2, August 2017, Pages 139-151, Flavored Tobacco Product Use in Youth and Adults: Findings From the First Wave of the PATH Study (2013–2014)

Young Adult Tobacco and E-cigarette Use Transitions: Examining Stability Using Multistate Modeling, Raymond Niaura, PhD, Ilan Rich, Amanda L Johnson, MHS, Andrea C Villanti, PhD, MPH, Alexa R Romberg, PhD, Elizabeth C Hair, PhD, Donna M Vallone, PhD, MPH, David B Abrams, PhD

Correlates of Transitions in Tobacco Product Use by U.S. Adult Tobacco Users between 2013–2014 and 2014–2015: Findings from the PATH Study Wave 1 and Wave 2, International Journal of Environmental Research and Public Health

The other reviewer was correct to be concerned about unevenness across the data sets. In the section about sample datasets, it would make sense to highlight the similarities and differences after the datasets are introduced.

For each of the measures, it would be helpful to always have a justification for why the data was treated in the way that it was and likely a citation. For example, why was the cutoff for the highest smoking category 22 days? The doesn’t really make good conceptual sense (e.g. daily smokers may be different than everyone else) and I don’t understand why that was chosen based on the literature.

To continue along this line, for the network measures, provide citations for places where these measures have been used in similar research as a way to justify the choice of these measures.

I also think the updating that is necessary in the introduction should be followed by changes in the discussion.

7. PLOS authors have the option to publish the peer review history of their article (what does this mean?). If published, this will include your full peer review and any attached files.

Reviewer #1: No

Reviewer #2: No

---

## [Author Response · Author response to Decision Letter 1]

7 Feb 2021

Response to Reviewers

PONE-D-20-32056R1

Smoking prevalence, core/periphery network positions, and peer influence: findings from five datasets on US adolescents

PLOS ONE

I would like to thank the editor and reviewer for their very constructive feedback. Incorporating the reviewer’s suggestions for revision has resulted in a greatly improved manuscript. Below, I note the concerns of the reviewer and then explain how I responded to each comment. 

1. I appreciate the citations in the methods about adolescence. However, tobacco control researchers, who will be reading this if it is published, often treat individuals 18-24 as separate from adolescents. It’s fine to include older young adults in this analysis and I understand the NIH and FDA and other organizations treat them as children but your audience will not. This paper will be more credible to your audience if you engage with the norms they use the talk about this population. It would not be hard to reframe the literature review. This would also give you the opportunity to incorporate, more broadly, relevant literature on smoking behavior in your population of interest. You could incorporate this literature and then say that you are going to use the term adolescent but this young adult population is of specific interest in tobacco control and other areas. Please see the following citations:

(1) Smoking impacts on prefrontal attentional network function in young adult brains, Francesco Musso, Franziska Bettermann, Goran Vucurevic, Peter Stoeter, Andreas Konrad & Georg Winterer, Psychopharmacology volume 191, pages 159–169(2007)

(2) American Journal of Preventive Medicine, Volume 53, Issue 2, August 2017, Pages 139-151, Flavored Tobacco Product Use in Youth and Adults: Findings from the First Wave of the PATH Study (2013–2014)

(3) Young Adult Tobacco and E-cigarette Use Transitions: Examining Stability Using Multistate Modeling, Raymond Niaura, PhD, Ilan Rich, Amanda L Johnson, MHS, Andrea C Villanti, PhD, MPH, Alexa R Romberg, PhD, Elizabeth C Hair, PhD, Donna M Vallone, PhD, MPH, David B Abrams, PhD

(4) Correlates of Transitions in Tobacco Product Use by U.S. Adult Tobacco Users between 2013–2014 and 2014–2015: Findings from the PATH Study Wave 1 and Wave 2, International Journal of Environmental Research and Public Health

––––––––––––––––––––––––––––––––––––––––––––––––––––––––––––––––––––––

I thank the reviewer for this suggestion. I now incorporate the line of literature on young adult smokers, as shown in the paragraph on pages 6-7 in the revised manuscript.

“Before proceeding to the next section, we want to note that researchers from various disciplines may have different definitions of the term adolescent. For example, the US Food and Drug Administration (FDA) defines adolescents as individuals in the 12-21 years age group (FDA 2003) and American Academy of Pediatrics (AAP) identifies adolescence as 11 to 21 years of age, including early (ages 11-14 years), middle (ages 15-17 years), and late (ages 18-21 years) adolescence (Hagan et al. 2008). Tobacco control researchers, on the other hand, often treat individuals in the 18-24 years age group as young adults who are separated from adolescents (Musso et al. 2007; Villanti et al. 2017; Kasza et al. 2018; Niaura et al. 2020). Therefore, we incorporate these perspectives from multiple disciplines: we use the term adolescent throughout this study, but we also cover young adult population who are of specific interest in tobacco control and relevant areas.”

In the “Results” section, after introducing the age composition of subjects in the five sample datasets on pages 11-12, I add “the participants in the five sample datasets fit the definitions of adolescents by FDA and AAP (FDA 2003; Hagan et al. 2008), and some of them are also among the young adult population of specific interest in tobacco control and relevant areas (Musso et al. 2007; Villanti et al. 2017; Kasza et al. 2018; Niaura et al. 2020).”

2. The other reviewer was correct to be concerned about unevenness across the data sets. In the section about sample datasets, it would make sense to highlight the similarities and differences after the datasets are introduced.

––––––––––––––––––––––––––––––––––––––––––––––––––––––––––––––––––––––

As the reviewer has suggested, I add a paragraph after introducing the datasets to highlight their similarities and differences on pages 8-9 in the revised manuscript:

“All five sample datasets contain the information on their participants’ self-reported personal characteristics (i.e., gender, age, race, parental education level) and smoking levels at two consecutive time points. There are also nonnegligible differences across the datasets. First, the first three sample datasets were collected from high school students during their middle and late adolescent years, while sample datasets 4-5 were collected from college students in late adolescence and emerging adulthood. Second, the NetSense and NetHealth projects used a different strategy from the Add Health study to collect information on friendship networks. The Add Health study used the “name generator”, i.e., a survey question asking each participant to recall the names of his or her friends. Instead, the NetSense and NetHealth projects built friendship networks among their participants with real-word communication events unobtrusively collected by their smartphones. Third, the Zeitgeist toward smoking in 1990s, when the first three sample datasets were collected, was different from that in 2010s, when sample datasets 4-5 were collected.”

3. For each of the measures, it would be helpful to always have a justification for why the data was treated in the way that it was and likely a citation. For example, why was the cutoff for the highest smoking category 22 days? The doesn’t really make good conceptual sense (e.g. daily smokers may be different than everyone else) and I don’t understand why that was chosen based on the literature.

––––––––––––––––––––––––––––––––––––––––––––––––––––––––––––––––––––––

This is a great question. I used the same categories as defined in Lakon et al. (2014), Wang et al. (2016), and Wang et al. (2018), and these citations are now added as a justification on page 9 in revised manuscript. 

4. To continue along this line, for the network measures, provide citations for places where these measures have been used in similar research as a way to justify the choice of these measures.

––––––––––––––––––––––––––––––––––––––––––––––––––––––––––––––––––––––

This is a valid point. Degree centrality and eigenvector centrality have been used as indicators of core-periphery positions in previous research on European tourist online networks (David‑Negre et al. 2018) and international investment networks (Galaso et al. 2020). Betweenness centrality is found to be highly associated with peer influence in crime co-offending networks (Bouchard & Konarski 2014) and online social networks (Johnson et al. 2015). k-core has been used as a coreness measure to study HIV risk (Rice et al. 2012; Shahesmaeili et al. 2015) and COVID-19 immunity (Douglas et al. 2020). These citations are now added as a justification of network measures on pages 9-10 in revised manuscript. 

5. I also think the updating that is necessary in the introduction should be followed by changes in the discussion.

––––––––––––––––––––––––––––––––––––––––––––––––––––––––––––––––––––––

I revise all “adolescent smokers” to “adolescent and young adult smokers” in the “Discussion” section and make updates as the reviewer has suggested. 

References

Bouchard M, Konarski R. Assessing the core membership of a youth gang from its co-offending network. In: Morselli C, editor. Crime and Networks. New York, NY: Routledge; 2014. pp. 81–96.

David-Negre T, Almedida-Santana A, Hernández JM, Moreno-Gil S. (2018). Understanding European tourists’ use of e-tourism platforms: Analysis of networks. Inf Technol Tour. 2018; 20: 131–152.

Douglas PK, Farahani FV, Douglas DB, Bookheimer, S. (2020). Convalescent blood treatment for COVID-19: Are local donors enough? arXiv 2009.12773 [Preprint]. 2020. Available from: https://arxiv.org/abs/2009.12773.

Galaso P, Sanchez-Diez, A. Core-periphery relations in the international mergers and acquisitions network. Appl Econ Int Dev. 2020; 20: 23–34.

Hagan JF Jr, Shaw JS, Duncan P, editors. Bright Futures: Guidelines for the Health Supervision of Infants, Children, and Adolescents. 3rd ed. Elk Grove Village, IL: American Academy of Pediatrics; 2008.

Johnson SL, Safadi H, Faraj S. The emergence of online community leadership. Inf Syst Res. 2015; 26: 165–187.

Kasza KA, Coleman B, Sharma E, Conway KP, Cummings KM, Goniewicz ML, et al. Correlates of transitions in tobacco product use by US adult tobacco users between 2013–2014 and 2014–2015: Findings from the PATH study wave 1 and wave 2. Int J Environ Res Public Health. 2018; 15: 2556.

Lakon CM, Wang C, Butts CT, Jose R, Timberlake DS, Hipp JR. A dynamic model of adolescent friendship networks, parental influences, and smoking. J Youth Adolesc. 2014; 44: 1767–1786.

Musso F, Bettermann F, Vucurevic G, Stoeter P, Konrad A, Winterer G. Smoking impacts on prefrontal attentional network function in young adult brains. Psychopharmacology. 2007; 191: 159–169.

Niaura R, Rich I, Johnson AL, Villanti AC, Romberg AR, Hair EC, et al. Young adult tobacco and e-cigarette use transitions: Examining stability using multistate modeling. Nicotine Tob Res. 2020; 22: 647–654.

Rice E, Barman-Adhikari A, Milburn NG, Monro W. Position-specific HIV risk in a large network of homeless youths. Am J Public Health. 2012; 102: 141–147.

Shahesmaeili A, Haghdoost AA, Soori H. Network location and risk of human immunodeficiency virus transmission among injecting drug users: Results of multiple membership multilevel modeling of social networks. Addict Health; 2015; 7: 1–13.

US Food and Drug Administration (FDA). Guidance for Industry and FDA Staff: Pediatric Expertise for Advisory Panels. Rockville, MD: US Department of Health and Human Services, Food and Drug Administration, Center for Devices and Radiological Health; 2003. Available from: https://www.fda.gov/media/72451/download.

Villanti AC, Johnson AL, Ambrose BK, Cummings KM, Stanton CA, Rose SW, et al. Flavored tobacco product use in youth and adults: Findings from the first wave of the PATH study (2013–2014). Am J Prev Med. 2017; 53: 139–151.

Wang C, Hipp JR, Butts CT, Jose R, Lakon CM. Coevolution of adolescent friendship networks and smoking and drinking behaviors with consideration of parental influence. Psychol Addict Behav. 2016; 30: 312–324.

Wang C, Hipp JR, Butts CT, Lakon CM. The interdependence of cigarette, alcohol, and marijuana use in the context of school-based social networks. PLoS ONE. 2018; 13: e0200904.

---

## [Decision Letter · Decision Letter 2]

4 Mar 2021

PONE-D-20-32056R2

Smoking prevalence, core/periphery network positions, and peer influence: findings from five datasets on US adolescents

PLOS ONE

Dear Dr. Wang,

Thank you for submitting your manuscript to PLOS ONE. After careful consideration, we feel that it has merit but does not fully meet PLOS ONE’s publication criteria as it currently stands. Therefore, we invite you to submit a revised version of the manuscript that addresses the points raised during the review process.

Please revise the paper to make it clear that you are studying young adults and not adolescents.  There has been a move at NIH, for example, to stop designating people 18-21 as children.  The way that study population is studies will make your paper hard for people to use in the field.

We look forward to receiving your revised manuscript.

Kind regards,

Stanton A. Glantz

Academic Editor

PLOS ONE

Journal Requirements:

Reviewers' comments:

Reviewer's Responses to Questions

**Comments to the Author**

1. If the authors have adequately addressed your comments raised in a previous round of review and you feel that this manuscript is now acceptable for publication, you may indicate that here to bypass the “Comments to the Author” section, enter your conflict of interest statement in the “Confidential to Editor” section, and submit your "Accept" recommendation.

Reviewer #2: All comments have been addressed

2. Is the manuscript technically sound, and do the data support the conclusions?

Reviewer #2: Yes

3. Has the statistical analysis been performed appropriately and rigorously? 

Reviewer #2: Yes

4. Have the authors made all data underlying the findings in their manuscript fully available?

Reviewer #2: Yes

5. Is the manuscript presented in an intelligible fashion and written in standard English?

Reviewer #2: Yes

6. Review Comments to the Author

Reviewer #2: I wish that you would consider my point that you are studying young adults and not adolescents. There has been a move at NIH, for example, to stop designating people 18-21 as children. I believe this will make your paper hard for people to use in the field. However, you have addressed my comments.

7. PLOS authors have the option to publish the peer review history of their article (what does this mean?). If published, this will include your full peer review and any attached files.

Reviewer #2: No

---

## [Author Response · Author response to Decision Letter 2]

8 Mar 2021

Response to Reviewers

PONE-D-20-32056R2

Smoking prevalence, core/periphery network positions, and peer influence: findings from five datasets on US adolescents

PLOS ONE

I would like to thank the editor and reviewer for their very constructive feedback. Incorporating the editor and reviewer’s suggestions for revision has resulted in a greatly improved manuscript. Below, I note the concerns of the editor and then explain how I responded to each comment. 

Editor

Please revise the paper to make it clear that you are studying young adults and not adolescents. There has been a move at NIH, for example, to stop designating people 18-21 as children. The way that study population is studies will make your paper hard for people to use in the field.

––––––––––––––––––––––––––––––––––––––––––––––––––––

I thank the editor for this suggestion. In the revised manuscript I completely adopt the tobacco control researchers’ approach by defining individuals in the 18-24 years age group as young adults who are separated from adolescents aged 12-17. The participants from the sample datasets 4 and 5 had an average age of 18.28 and 18.01, respectively, and thus fit this definition. However, the participants from the first three sample datasets had an average age of 15.53, 15.95, and 14.19, respectively. The readers will be utterly confused if I call them young adults as well. As a result, in the revised manuscript I clarify I am studying both adolescents and young adults, and corresponding changes are made in the title, abstract, and manuscript.

---

## [Editor Report · Decision Letter 3]

10 Mar 2021

Smoking prevalence, core/periphery network positions, and peer influence: findings from five datasets on US adolescents and young adults

PONE-D-20-32056R3

Dear Dr. Wang,

We’re pleased to inform you that your manuscript has been judged scientifically suitable for publication and will be formally accepted for publication once it meets all outstanding technical requirements.

Kind regards,

Stanton A. Glantz

Academic Editor

PLOS ONE
---

## [Editor Report · Acceptance letter]

15 Mar 2021

PONE-D-20-32056R3 

Smoking prevalence, core/periphery network positions, and peer influence: findings from five datasets on US adolescents and young adults 

Dear Dr. Wang:

I'm pleased to inform you that your manuscript has been deemed suitable for publication in PLOS ONE. Congratulations! Your manuscript is now with our production department. 

Kind regards, 

on behalf of

Professor Stanton A. Glantz 

Academic Editor

PLOS ONE